# A comprehensive but practical methodology for selecting biological indicators for long-term monitoring

Roger Puig-Gironès[1,2]*, Joan Real[1]

1 Equip de Biologia de la Conservació, Departament de Biologia Evolutiva, Ecologia i Ciències Ambientals & Institut de la Recerca de la Biodiversitat (IRBIO), Universitat de Barcelona, Barcelona, Catalonia, Spain, 2 Departament de Ciències Ambientals, University of Girona, Girona, Catalonia, Spain

* rogerpuiggirones@gmail.com

## Abstract

The selection of the many biological indicators described in scientific literature is rarely based on systematic or clear-cut processes, and often takes into account only a single or very few taxa, or even disregards the complex interactions that exist between the components of biodiversity. In certain cases, the particular context of a site–for example in the Mediterranean Basin–makes it difficult to apply the choice of indicators to other regions proposed in the literature. Therefore, the selection of appropriate methodologies for generating relevant indicators for a particular site is of crucial importance. Here, we present a simple quantitative methodology capable of incorporating multidisciplinary information for assessing and selecting appropriate methods and indicators for monitoring local biodiversity. The methodology combines several ecological levels (species, habitats, processes, and ecosystem disturbances), and embraces biological interactions and common functional guilds (detritivores, producers, herbivores, and carnivores). We followed an iterative selection procedure consisting of five phases: 1) collection focal area useful information; 2) classification of this information into interrelated datasets; 3) assessment and selection of the relevant components using a quantitative relevance index; 4) the adding of taxonomic, physiognomic and functional similarities to the relevant components; and 5) the quantitative selection of the priority indicators in the study area. To demonstrate the potential of this methodology, we took as a case study the biodiversity components and their ecological interactions present in a protected area. We show that our methodology can help select appropriate local and long-term indicators, reduce the number of components required for thorough biodiversity monitoring, and underline the importance of ecological processes.

## 1. Introduction

Ecosystems have many different biological components, a fact that hinders any attempt to attain knowledge of the entirety of the elements that constitute our natural biodiversity [1]. At the ecological level, the elements of biodiversity are organized into complex networks that

**Data Availability Statement:** As the underlying data contains information about endemic and endangered species, it cannot be made publicly available. For requirement of data available contact

with Sant Llorenç del Munt i l'Obac Natural Parc managers (p.santllorenc@diba.cat).

**Funding:** This work was supported by the Diputació de Barcelona, Universitat de Barcelona, Fundació Bosch i Gimpera and Fundación de la Biodiversidad; however the funders had no role in study design, data collection and analysis, decision to publish, or preparation of the manuscript.

**Competing interests:** The authors have declared that no competing interests exist.

interact with biotic and abiotic components in ecosystem processes and produce ecosystem services for use by human societies [2]. Therefore, understanding the causes and consequences of the current loss of biological components is fundamental [3, 4]. The availability of useful information regarding the state and trends operating in our biodiversity is a crucial step in the construction of biological indicators [5]. These indicators interact with the ecosystem and reflect the changes occurring in a habitat, community, or ecosystem; they provide information about complex ecological processes, act as early warning signals, help diagnose the cause of ecological problems, and are important tools for use in conservation planning and management [6–10].

Understanding the trends and drivers of biodiversity change is vital when attempting to decide on appropriate conservation measures [11, 12]; nevertheless, to do so requires robust and comprehensive information obtained from biodiversity monitoring programs [13]. Despite the enormous challenges facing global biodiversity conservation [14], it is essential to quantify and predict local and regional variations to be able to address and protect all aspects of biodiversity. The Group on Earth Observations Biodiversity Observation Network (GEO BON) has developed the concept of Essential Biodiversity Variables (EBVs) [15], which can be used to link local and global needs and aims [16]. Conceptually, EBVs are located between primary data observation and indicators [15, 17], and have been developed to help prioritize a minimum set of essential measures for the consistent study, reporting, and management of all the major elements of biodiversity change. GEO BON has proposed 22 candidate variables belonging to six EBV classes: genetic composition, species populations, species traits, community composition, ecosystem functioning, and ecosystem structure [16]. Some EBVs such as population abundance–a continuous variable for all taxa–may be difficult to obtain due to the large number of taxa found at a single site. Thus, it may be more advantageous in a particular region to select just a few local biodiversity components as candidates for monitoring and it follows that standardised local sampling at fine resolutions and the datasets it generates will become a necessary and useful part of the development of global biodiversity indicators [14]. These specific monitoring systems will provide accurate information for creating local indicators for decision-making that, at the same time, can be incorporated into global EBVs [16].

Despite the extensive scientific literature that exists on the selection of indicators, this process is often neither systematic nor methodical [1, 18–22]. The selection criteria for indicators may be related to the distribution, abundance, richness, functional importance, or sensitivity of taxa to environmental change [15, 20, 23–28]. However, the choice of indicators is usually based on previously cited research, the conservation status of taxa, and/or the ease with which data can be sampled, sorted, and identified [29]. Furthermore, choices may even be based on subjective criteria unrelated to ecological criteria [20, 30, 31] or be driven by the availability of data [32]. Consequently, given the huge number of taxa that meet these requirements [33], a multitude of indicators have been described in the literature [29]; thus, when selecting appropriate indicators their relation to the local context and ecosystems must be taken into account. Accordingly, the knowledge of experts or specialists in local taxa is essential since one aspect of biodiversity (species, habitats, ecological processes, and biotic, abiotic, and anthropic problems) may affect a focal region differently and so require its indicator [6]. It is also important to assess which indicators are valid and informative for a region and which are redundant, overvalued, or unnecessary.

The Mediterranean Basin is a biodiversity hotspot that, due to the unique ecological processes and heterogeneous climatic conditions that drive its ecosystems, harbours numerous endemic plant and animal species [34–37]. Moreover, this region is experiencing a multitude of environmental impacts related to the great anthropic presence (e.g. urbanisation, infrastructures, resource overexploitation, or frequent wildfires) that complicate the study, monitoring,

and biodiversity predictions of its vast number of multidirectional ecological relationships. Hence, a suitable selection of indicators is as essential as is the correct design of monitoring protocols with specific objectives. In Mediterranean ecosystems, objective methods for selecting an appropriate set of biodiversity indicators are lacking, or are limited to just one or a few groups [38, 39], and assessments often cannot be contrasted or are difficult to put into practice [40]. In other regions protocols and selection methods already exist and these methodologies generate a large variety of context-specific indicators [20, 28, 41–46].

Our overall aim was thus to develop a simple quantitative methodology capable of incorporating multidisciplinary information for assessing and selecting appropriate methods and indicators for monitoring local biodiversity. This methodology combines several ecological levels (species, habitats, processes, and ecosystem disturbances [1]), and embraces biological interactions and common functional guilds (producers, herbivores, carnivores, and detritivores [47]). We applied our methodology to Sant Llorenç del Munt i l'Obac Natural Park (NE Spain), a protected area possessing an important and representative range of Mediterranean ecosystems. Despite focussing on Mediterranean areas, our approach is relevant to other regions, landscapes and ecosystems in which there are similar challenges to biodiversity conservation.

## 2. Materials and methods

### 2.1. Methodology approach

We developed a multi-criteria [48] methodology to select relevant biodiversity components (species, habitats, ecological processes, and ecosystem disturbances), and then used these components to create biodiversity indicators for monitoring biodiversity in particular sites. We followed a hierarchical selection procedure (1) so that the resulting indicators combining trends in diversity, reproductive success, and growth rates would be effective in detecting ecological changes and useful for assessing management impacts [29, 49, 50]. The process of selection of priority indicators to implement in long-term monitoring biodiversity following five steps: 1) the collection of published and unpublished information on the species, habitats, and biological communities present in the action scope; (2) the classification of the components of biodiversity in candidates of species, habitats, and ecosystem disturbances; (3) the establishment of the relevant components of species, habitats, ecological processes, and ecosystem disturbances employing the assessment with an Index of Relevance focusing on ecological networks; (4) the creation of a Monitoring Catalogue of relevant components grouped by similarity; and (5) the establishment of the *Priority indicators* by a quantitative Priority index.

**2.1.1. State 1: Collection of available information.** The primary components of the biodiversity (i.e. species and habitats) cited in the focal area were identified and collated. Subsequently, information regarding the stressors and drivers–i.e. the specific functional, compositional, and structural components or ecosystem disturbances such as natural and anthropogenic stressors [51]–of these biodiversity components was compiled (Fig 1).

**2.1.2. State 2: Classification of the components of biodiversity in datasets of candidates of species, habitats, and ecosystem disturbances.** Based on a literature review, the collected information was classified into different datasets (Fig 1). To ensure a precise and rigorous selection, we divided the initial components of biodiversity into three ecological levels: species, habitats, and ecosystem disturbances. These datasets incorporate different variables affecting the stressors and drivers of the elements used to assess and select the relevant components. An effective collection of prior information on the area to be applied, especially not published reports, is essential. If relevant prior information is not available, you need to back to State 1 to search for or generate relevant information.

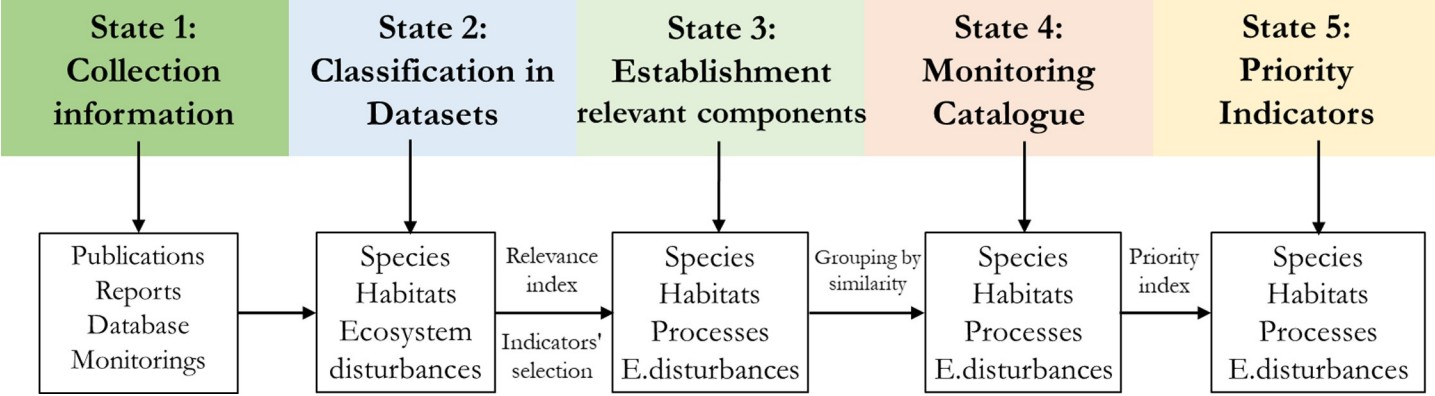

**Fig 1. The work schedule for select priority indicators.** The work schedule carried out for the selection of Priority indicators and its long-term monitoring program respects the following states: (1) search of published or unpublished information on the species, habitats, and biological communities present in the action scope; (2) classification of the components of biodiversity in candidates of species, habitats, and ecosystem disturbances; (3) establishment of and Relevant components catalogue of species, habitats, ecological processes and ecosystem disturbances employing the assessment with an Index of relevance, and make an ecological network to determine the Relevant components of the local ecological process; (4) creation of a Monitoring Catalogue of candidates; and (5) establishment of the *Priority indicators* by a quantitative Priority index.

**2.1.3. State 3: The establishment of the relevant components of monitoring biodiversity.** *2.1.3.1. Quantitative method.* To select the relevant components from the datasets we developed a quantitative Relevance Index focusing on different variables specified and related to species, habitats, and ecological processes (Table 1), that in turn each one was subdivided into different levels and criteria (Table 2).

- For species, the quantitative Relevance Index value lies in the range 0 to 8 (Table 1) and selects all values with a threshold of $\geq 4$. The species' assessment parameters are related to the degree of threat [52], their ecological interest [53, 54], and the expert criterion [6] for each taxonomic group (Table 2). The ecological interest parameter gives priority to species that occupy a wide-ranging habitat, have rich habitat requirements (patch size, structure, and configuration), and depend on particular ecological processes (Table 2), given that species restricted to fewer habitat types are more susceptible to local impacts [55].

- At habitat level, the quantitative Relevance Index value is established from 0 to 8 (Table 1) and selects values with a threshold of $\geq 4$. For each habitat catalogued, the parameters are the degree of threat, ecological interest, the representativeness of preselected species that depend on each one of the assessed habitats, and the expert criterion (Table 2).

- In the case of ecosystem disturbances, the Relevance Index runs from 0 to 6 (Table 1), and, once assessed, values with a threshold of $\geq 3$ are selected. In this case, the parameters used

**Table 1. The quantitative scoring ranges and the relevance index formula used in our multi-criteria analysis for the relevant candidates of species, habitats, and ecosystem disturbances.**

|  | Species | Habitats | Ecosystem disturbances |
|---|---|---|---|
| **a) Degree of threat** | 0 to 3 | 0 to 2 |  |
| **b) Ecological interest** | 0 to 3 | 0 to 2 |  |
| **c) Representativeness** |  | 0 to 2 | 0 to 2 |
| **d) Habitats** |  |  | 0 to 2 |
| **e) Expert or specialist criterion** | 0 to 2 | 0 to 2 | 0 to 2 |
| **Relevance index** | a + b + e | a + b + c + e | c + d + e |
|  | = max. 8 | = max. 8 | = max. 6 |

**Table 2. Summary of the assessment parameters used to select the relevant candidates, the ecological level (species, habitat, or ecosystem disturbances), the valuation for the calculation of the relevance index, the parameter's description, and the specific criterion to assess each of the parameters.**

| Assessment parameters | Level | Value | Description | Criterion |
|---|---|---|---|---|
| Degree of threat | Species and habitats | 0 to 1 | Absence/presence on local plans or reports. | A value of 1 is assigned when the species or taxon appeared in one or more strategic reports or management plans at the local scale. |
| | Species | 0 to 1 | Absence/presence on current legislation. | When the species appears in the current legislation or annexe thereof, at the level of the autonomous community, country, or European community, it receives a value of 1. |
| | Species | 0 to 1 | Cataloguing on the IUCN Red List. | When the species is catalogued (vulnerable, endangered, or critically endangered) for the IUCN Red List at the country level, it is assigned a value of 1. |
| | Habitat | 0 to 1 | Cataloguing on European Directive | When the habitat is a priority or of interest for the European directive, it receives a value of 1. |
| Ecological interest | Species | 0 to 2 (if fulfil a criterion = 1; if accomplish more than one = 2) | Habitat specialization | Species representative of a particular habitat, especially of rare habitats. |
| | | | Geographic distribution | Species with a disjoint geographical distribution (broad geographical separation between populations) |
| | | | Effects of climate | Species with their distribution boundary in the area of study and, therefore, the variations in the climate can affect it. |
| | | | Ecological process | Species relevant in some ecological processes (pollination, herbivorism, production of trophic resources, predator-prey relationship, seed dispersion, parasitism, etc.) |
| | | | Ecosystem disturbances | Species considered ecological indicators of environmental quality, water quality, unsustainable management, being affected by forest pests, being a hunting object, invasive/allochthon species, abundant in undisturbed areas, affected by human frequentation, etc. |
| | | | Other aspects | Other relevant aspects of ecological interest such as the specialized diet, rarity or symbolism of the species, short-term population trends, etc. |
| | | 0 to 1 | Absence/presence on monitoring programs | If it fulfils, the species obtain a value of 1 in this section as they correspond to common species (commonly detected species in monitoring plans). |
| | Habitat | 0 to 2 (if fulfil a criterion = 1; if accomplish more than one = 2) | Abundance | Habitats with a considerable extension in the area of study. |
| | | | Singularity | Rare, regressive, or poorly represented habitat in the area of study. |
| | | | Ecological processes | Habitats in which important ecological processes occur for the global functioning of ecosystems, e.g. production of trophic resources, herbivorism, etc. |
| | | | Ecosystem disturbances | Habitats that are more susceptible to suffer relevant ecosystem disturbances, such as forest exploitation, afforestation, human frequentation, wildfires, etc. |
| | | | Other aspects | Other important or relevant characteristics of the habitats. |
| Representativeness | Habitat and Ecosystem disturbances | 0 to 2 | Number of affected species | Those habitats with a greater number of species or are affected by the ecosystem disturbances, have a higher value. The total number of species is relocated over a value of 2 |
| Habitats | Ecosystem disturbances | 0 to 2 | Number of affected habitats | Ecosystem disturbances that affect a greater number of habitats present a higher value. The total number of habitats is relocated over a value of 2. |
| Expert or specialist criterion | Species, Habitat and Ecosystem disturbances | 0 to 2 | Local value awarded by scientific experts | The criterion of external expert or taxon specialist grants a value of 2 to the species, habitats, and/or ecosystem disturbances with high importance and relevance in the context of the study area, and 1 to species, habitats, and/or ecosystem disturbances with relative importance in the study area. |

were related to the species and habitats potentially affected by each ecosystem disturbances, and to expert criteria (Table 2).

Once the species, habitats, and ecosystem disturbances have been assessed, those that reach the threshold value of the Relevance Index are selected. Because the threshold value needs to be sufficiently robust but at the same time to include potential under-detected components, we established as a criterion to maintain the components with a Relevance index equal to or above half of the maximum global potential value that included valuable ecosystem components, as we have verified in our previous tests (see Table 1).

For the component assessment for species, habitats, and ecosystem disturbances accurate knowledge of ecological requirements and responses to environmental changes in the candidates is required when choosing the elements that reflect our ultimate aim. Here, the bibliography and expert criteria are essential (Table 2), and will help highlight species or habitats that are not present or not evaluated as threatened. The ecological interest contribute to minimize the weight of these threatened and singular component, which do not necessarily reflect the spatial and temporal trends of biodiversity [23, 56] but, rather, respond to the local conservation status [57].

*2.1.3.2. Construct an interrelated diagram to determine the ecological processes.* Ecological processes were qualitatively assessed strictly on preselected relevant candidates (species, habitats, and ecosystem disturbances) and their diversity of interactions in the focal area, i.e. the diversity and structure of multi-trophic interactions between organisms and habitat types. These processes can be complex to define and elucidate and their monitoring is often difficult to perform. Thus, basic processes such as trophic relationships and mutualism interactions are required to select the processes [46]. Here, based on the previously relevant candidates, a diagram is established that depicts all possible direct and indirect trophic relationships, as well as the interactions between species, habitats, and external ecosystem disturbances (abiotic and anthropic). From here, the main ecological processes in the study area are defined as the water cycle, nutrient cycle, ecological succession, trophic networks, mutualisms, and other local relevant processes (Fig 2). This exercise forces to define the biological and geographical limits (monitor scale) and, then, make practical decisions about how much can be done. In addition, it emphasises that to monitor any ecosystem you must have a great deal of background data [46].

**2.1.4. State 4: Monitoring catalogue.** Once the different relevant components have been assessed and selected, a Monitoring Catalogue (Fig 1) can be established to act as a strategic guide, facilitate understanding, and identify which variables need to be sampled and measured to generate the biological indicators in an ecological sense and to avoid redundancies. To attain the list of relevant components, (1) we grouped components by taxonomic, physiognomic, and functional similarities (e.g. common birds, freshwater invertebrates, decomposers). Then, (2) classified the grouped relevant components in four ecological levels: species, habitat, ecological process, and ecosystem disturbances (see the case study below as a practical example). (3) For each group of relevant components we selected the adequate variables to be sampled, e.g., species presence and abundance, reproductive and survival taxes, habitat preferences, soil structure, and composition. Lastly, (4) incorporated into each level all the monitoring components whilst bearing in mind that each can be part of one or more levels depending on their role and interactions within the ecosystem (see State 3). For example, species abundance and richness of butterflies provide information about the species level and their community respectively, so in the latter case indicate the variation on ecosystem functional capabilities (e.g. as primary consumers), climate change or afforestation effects.

**2.1.5. State 5: Priority indicators.** Priority Indicators are useful for acquiring knowledge of biological and ecological changes in a given area and indispensable to optimize the efforts in

**Fig 2. Interactions between the monitoring candidates to elucidate ecological processes.** Diagram of the possible interactions that are established between the monitoring candidates, where highlight five main groups of organisms in the trophic network, the detritivores (green box), the primary producers, the primary and secondary consumers, and the predators. However, within these groups, interactions occur, even within the same species communities. On the other hand, this entire complex trophic network is conditioned by abiotic and anthropogenic external ecosystem disturbances (ecosystem disturbances; red box), such as climatology, perturbations, pollution, forest management, etc. Continuous lines denote direct relationships, while discontinuous lines describe diffused relationships often characteristic of opportunistic species.

monitoring. In our scheme, they were selected using a quantitative Priority Index, which awards importance from 0 to 5 based on the criteria of specificity (key species or habitats, etc.), generality (abundant species, representative and common processes, etc.), importance (endemic species, determining ecosystem disturbances, etc.), interactions between organisms, and usefulness for management and decision-making (Table 3). In this case, only those Monitoring Components that reach maximum importance (Priority Index $\geq$ 4) are chosen as Priority Indicators in the study area.

Priority Indicators were divided into four interrelated ecological levels:

a. Species. The indicators of the state of species, in general, are designed to identify the population and conservation status of the most characteristic species in ecosystems, whose presence or abundance indicates population trends and the state of the habitats and ecosystems in which they live.

b. Habitats. The indicators of the state of habitats are based on knowledge of the condition of plant communities (floristically and functionally) and land use, and, in particular, of the evolution of their structural, spatial, and temporal evolution and distribution.

c. Ecological processes. The indicators highlight the relationships that link organisms or groups of organisms with each other and with the environment (habitat or ecosystem) that hosts them and the specific interactions between them.

**Table 3. Summary of the assessment parameters used to select the priority indicators by a quantitative priority index using five parameters with its specific criterion.**

| Parameter | Value | Criterion |
|---|---|---|
| **Specificity** | 0 to 1 | A value of 1 is assigned when the indicator allows capturing the tendencies and dynamics of the species, communities, or habitats and relevant ecological processes or ecosystem disturbances, in the study area. |
| **Generality** | 0 to 1 | A value of 1 is assigned when the indicator is represented by abundant species or communities, representative habitats, common ecological processes, or relevant and extensive ecosystem disturbances, in the study area. |
| **Importance** | 0 to 1 | A value of 1 is assigned when the indicator involves the presence of singular species, unique habitats, key ecological processes, determining ecosystem disturbances, etc. |
| **Interactions between organisms** | 0 to 1 | A value of 1 is assigned when the indicator involves different organisms, communities, key ecological processes, are altered by relevant ecosystem disturbances, etc. |
| **Usefulness for management and decision making** | 0 to 1 | A value of 1 is assigned when the indicator involves components that can provide useful information for the management or conservation. |

d. Ecosystem disturbances. These indicators reveal the effects of the ecosystem disturbances that affect and alter the natural functioning of ecosystems and their constituent organisms. Disturbances in ecosystems may be the product of natural cycles or anthropogenic activity.

## 2.2. Pilot study area

To apply our quantitative and qualitative methodology for selecting biodiversity indicators at the local scale, Sant Llorenç del Munt i l'Obac Natural Park (henceforth PNSLL; Fig 3) was chosen as the focal region. This mountainous area forms part of the Catalan Prelitoral Mountain Range and has an altitudinal range of 280–1,100 m a.s.l. and a typical mid-altitude Mediterranean montane climate. Its orography and geographical situation afford it great climatic variability, with an annual mean rainfall of 500–800 mm and a mean annual temperature of 15 C˚. Its lithology consists of permeable conglomerates with an argillaceous and calcareous matrix [58]. It is characterized by rocky outcrops and cliffs [59], whose forested crags and ridges are covered chiefly by holm oak (*Quercus ilex* L.) forests and, in the more humid valleys, patches of deciduous pubescent (*Quercus pubescens* Willd.) and sessile (*Quercus petraea* (Matt.) Liebl) oak forests. Lower areas are dominated by Aleppo pine (*Pinus halepensis* Mill.) woodland with an evergreen oak understory [60], while black (*Pinus nigra* subsp. *salzmannii* (Dunal) Franco) and Scots (*Pinus sylvestris* L.) pine forests cover smaller areas of terrain. Shrubland is also present mainly as a result of the wildfires that occurred during the 2000s. Currently, this 13,694-ha protected area is subject to intense human pressure (e.g. wildfires, human frequentation, and forest exploitation) since is located near the Barcelona conurbation (4 million inhabitants), one of the largest metropolitan areas in southern Europe.

## 3. Results

### 3.1. Information collection and classification into datasets

In all, 387 literature sources were consulted (S1 Appendix), including 213 publications, 66 unpublished reports, 10 technical plans, 19 datasets, and 79 reports from monitoring programs, as well as unpublished data by experts and researchers. A total of 3,226 species were catalogued and assessed for the PNSLL as candidates as indicators, including 52 algae, 159 fungi,

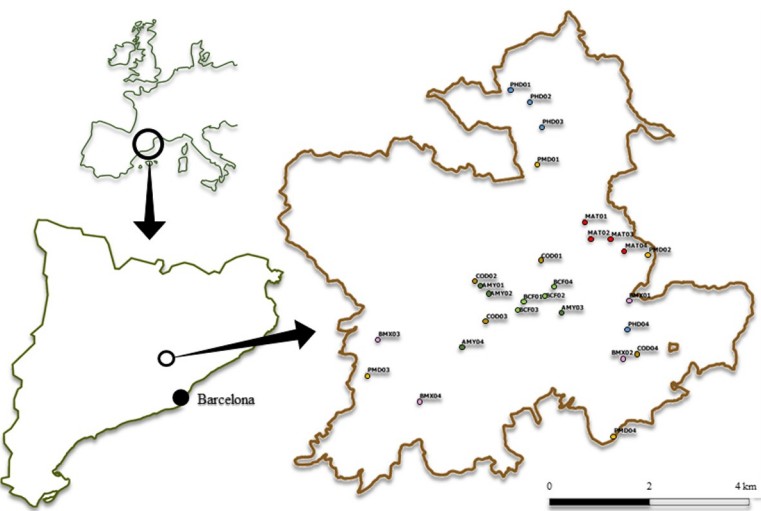

**Fig 3. Location of the pilot study area.** The geographic location of the Sant Lorenç del Munt i l'Obac Natural Park (white ring) in the nearby of the metropolitan area of Barcelona (black ring) in Catalonia (NE Iberian Peninsula) and its perimeter. Squares correspond to 100×100m Permanent Monitoring Plots, where COD correspond to Rocky areas habitat, MAT: Shrublands; PMD: Mediterranean pine forests, PHD: Wet pine forests; BMX: Mixed forests, AMY: Mountain holm oak forests, and BCF: Deciduous forests.

159 lichens, 69 bryophytes, 27 pteridophytes, 1,081 vascular plants (100 allochthones) [61], 1,404 invertebrates (1,179 insects, 53 gastropods, 139 arachnids, 30 from other groups and 7 allochthones), and 270 vertebrates (3 freshwater fish, 13 amphibians, 21 reptiles, 174 birds, 27 raptors, 51 mammals and 12 allochthones).

### 3.2. Relevant components selection

The relevant components were obtained through a Relevance Index which was calculated for all 3,226 assessed species (Table 4) of which 406 (12.6%) were selected as Relevant Components (S1 Table): plants contributed with 69 species (17%—of which 10 species were threatened species, 13 pteridophytes, 41 vascular plants, and five alien species), 191 species (47%) were invertebrates (of which 26 species were Odonata, 12 Gastropoda, 43 decomposers, 22 Formicidae, 30 Orthoptera; 36 Lepidoptera, ten other invertebrates of special interest, seven plague species and five were alien species), while 146 species (36%) were vertebrates (of which tree species were freshwater fish, seven amphibians, seven reptiles, 13 raptors, 70 common birds, 21 bats, seven small-mammals, two common medium-sized preys, two ungulates, five carnivores and nine were alien species).

**Table 4. Reduction in the number and percentage of candidates to relevant components through the use of the relevance index; and reduction of the number and percentage of monitoring components to priority indicators by a quantitative priority index.**

|  | Species | Habitats | Ecological processes | Ecosystem disturbances |
|---|---|---|---|---|
| **Candidates** | 3,226 | 96 | 8 | 14 |
| **Relevant components** | 408 | 11 | 6 | 9 |
| **Percentage of reduction** | -87.4% | -89.6% | -25.0% | -35.7% |
| **Monitoring components** | 17 | 2 | 5 | 8 |
| **Priority indicators** | 13 | 2 | 4 | 8 |
| **Percentage of reduction** | -23.5% | 0% | -20.0% | 0% |

We identified a total of 96 habitats in the PNSLL [62–65], which, to simplify the assessment and selection of habitats, were grouped according to floristic, structural, and ecological similarities, thereby generating 15 habitat categories to be assessed as candidates as indicators (S2 Table). Of the total 15 assessed habitats, eleven were selected (Table 4) as relevant components (using Relevance Index value; S3 Table): caves, rocky areas, dry meadows, shrublands, Mediterranean pine forests, humid pine forests, mixed forests, mountain holm oak forests, deciduous forests, riparian forests and freshwater habitats.

To identify the relevant ecosystem disturbances affecting Mediterranean ecosystem biodiversity and, in particular, our study area, an intense literature search was conducted. From the candidates' list of 14 ecosystem disturbances, nine were selected as Relevant Components (Table 4), corresponding to those with a Relevance Index value $\geq 3$ and including disturbances related to global change (climate change, wildfires, and alien species) and local ecosystem disturbances (afforestation, fragmentation, freshwater alteration, silvicultural activities, hunting, human frequentation, and fatalities on human infrastructures) (S4 Table).

At a global level, and using our quantitative method, from 3,344 candidates including species, habitats, and ecosystem disturbances a set of 434 Relevant Components were finally obtained. The main ecological processes operating in the study area were defined first drawing a diagram that illustrates trophic relationships (S1 Fig), as well as the interactions between species, habitats, and external ecosystem disturbances (abiotic and anthropic) resulting in decomposition, plant production, consumers and predators and mutualisms (pollination and zoochory). Other relevant ecological processes such as the water cycle or ecological succession were included in the ecosystem disturbances (e.g. climate change, alteration of aquatic ecosystems, wildfires, and afforestation; Table 5). Thus, the final number of relevant components was 434, an 87% decrease from the list of initial candidates (Table 4).

### 3.3. Monitoring catalogue and priority indicators

To constitute the Monitoring catalogue, we grouped the 434 Relevant components into independent levels of species, habitats, ecological processes, and ecosystem disturbances by similarities. In the case of 408 species were grouped by taxonomic, physiognomic, and functional similarities into 17 groups of species (threatened plant species, xeric gastropods, granivorous ants, decomposers, orthoptera, butterflies, freshwater macroinvertebrates, freshwater fishes, amphibians, reptiles, common birds, raptors, bats, small mammals, common medium-size preys, ungulates and carnivores). A total of 96 habitats were grouped in 11 habitat relevant categories (Table 4). Then, the habitat category resulted in two monitoring parameters (habitat structure and composition), which will be carried out in the 11 selected relevant habitats. Moreover, it is in these habitats where the specific transversal monitoring of species, processes and disturbances will be developed. Ecological processes were grouped by ecological functions in 6, decomposition, primary production, consumers, predators, zoocory, and pollination. Ecosystem disturbances were maintained in 8 (Table 4), climatic change, wildfires, afforestation, freshwater alterations, alien species, exploitation of natural resources, forests pests, human frequentation, human infrastructures). So resulted in a total of 32 specific Monitoring Components. Finally, to obtain the Priority Indicators we used the quantitative Priority Index, and 27 of these Monitoring Components were selected to monitor the biodiversity in the PNSLL (Table 5).

## 4. Discussion

Our methodology was established using a systematic and transparent conceptual framework. First, we divided our set of components into four interrelated ecological levels: species,

**Table 5. Priority indicators (n = 27) that quantitatively meet selection criteria on Sant Llorenç el Munt i l'Obac Natural Park.** The indicators are divided into four categories and in different attributes or pressures [1], consequently, each require different variables from monitoring candidates.

| Category | Attribute or pressure | Priority indicator | Variables from monitoring candidates |
|---|---|---|---|
| **State of the species** | **Dispersal-limited and special interest species** | Threatened flora | Species presence and abundance |
| | | | Flowering rate |
| | | | Occupied extension |
| | | Freshwater fishes | Species presence and abundance |
| | | Amphibian | Species presence and adult abundance |
| | | | Number of laying and larvae |
| | | Chiroptera | Species presence and abundance |
| | **Umbrella species** | Raptors | Species presence and abundance |
| | | | Reproductive and survival taxes |
| | **Link species** | Decomposers | Species presence and abundance |
| | | | Diversity |
| | | Orthoptera | Species presence and abundance |
| | | Small-mammals | Species presence and abundance |
| | | Common medium-size preys | Species presence and abundance |
| | | | Number of hunted individuals |
| | **Indicator species** | Butterflies | Species presence and abundance |
| | | | Habitat preferences |
| | | Freshwater macroinvertebrates | Species presence and abundance |
| | | | Diversity |
| | | Common birds | Species presence and abundance |
| | **Ecological engineers** | Ungulates | Species presence and abundance |
| | | | Number of hunted individuals |
| **State of the habitats** | **Landscape** | Structure | Habitat structure |
| | | | Regeneration |
| | | | Soil structure and composition |
| | | | Volume of necromass |
| | | Plant composition | Community composition |
| | | | Plant distribution |
| **State of the ecological processes** | **Ecosystem recourses** | Primary production | Vegetal production |
| | | | Mushroom production |
| | | | Acorn production |
| | | | Pinecone production |
| | **Trophic network** | Decomposition | Abundance and diversity of detritivores |
| | | | Volume of necromass |
| | | Consumers | Presence and abundance of orthopteran, small-mammals, common medium-size preys, common birds, and ungulates |
| | | Predators | Presence and abundance of carnivores and raptors |
| | | | Vital taxes |
| | | | Diet |
| **Ecosystem disturbances** | **Resistance and resilience** | Climatic change | Rainfall |
| | | | Temperature |
| | | | Relative humidity |
| | | | Insolation |
| | | | Evapotranspiration |
| | | | Extreme episodes |

(*Continued*)

**Table 5.** (Continued)

| Category | Attribute or pressure | Priority indicator | Variables from monitoring candidates |
|---|---|---|---|
| | | Wildfires | Presence and abundance of detritivores, ants, orthopteran, butterflies, reptiles, common birds, small-mammals and medium-size preys |
| | | | Plant composition |
| | | | Habitat structure |
| | | | Regeneration |
| | | | Soil erosion |
| | | | Trophic network |
| | | Afforestation | Presence and abundance of ants, orthopteran, reptiles, butterflies, common birds, and medium-size preys) |
| | | | Plant composition |
| | | | Habitat structure |
| | | Freshwater alterations | Flow level |
| | | | Contaminants and physical-chemical parameters |
| | | | Presence, abundance, and diversity of macroinvertebrates and freshwater fishes |
| | | | State of the riparian forest |
| **Biological invasions** | Alien species | | Species presence and abundance |
| | | | Distribution |
| **Anthropic pressure** | Exploitation of natural resources (silvicultural and hunting) | | Presence and abundance of detritivores, common birds, raptors, chiropters, medium-size preys, and ungulates |
| | | | Plant composition |
| | | | Habitat structure |
| | | | Soil erosion |
| | | | Trophic network |
| | | | Number of hunted individuals |
| | Human frequentation | | Presence and abundance of threatened flora, xeric gastropods, ants, orthopteran, butterflies, and common birds. |
| | | | Reproductive taxa of rocky and cavern species |
| | | | Plant composition |
| | | | Habitat structure |
| | | | Soil erosion |
| | | | Trophic networks |
| | | | Number of visitors |
| | Human infrastructures | | Roadkill taxa |
| | | | Risk and taxa of electrocution |
| | | | Risk and taxa of collision with power lines |
| | | | Fragmentation of river connectivity |
| | | | Alteration of fluvial flows |

habitats, ecological processes, and ecosystem disturbances. Species and habitat indicators provide information about specific states but other ecological levels are required if they are to be correctly interpreted since, for example, different bird species respond to specific habitat structure, the availability of resources, and environmental conditions [24, 38, 66]. Therefore, a single indicator cannot reveal ecosystem dynamics on its own, as all indicators are interlinked and their impacts are connected via a web of complex relationships (Fig 2).

In the components selection process quantitative criteria were given priority over more subjective qualitative criteria, and a multi-criteria analysis [48] was used to generate a set of candidates and assess these indicators in terms of their degrees of importance to the ecosystem. The

evaluation of each species, habitat, and ecosystem disturbance based on a points score using specific multi-criteria analysis allowed us to reduce the initial candidates to the final selected Priority Indicators (see State 3 in Methods). Exceptionally, we used qualitative selection in the particular case of ecological processes (see Discussion below). The use of a quantitative selection of indices also allowed us to drastically reduce the number of species and habitats to be monitored, which will thus optimise both future efforts and the use of resources. Some possible biases could arise from the exclusive use of the weight values in the Relevance Index as some potential candidates (e.g. plants and invertebrate species) would be undervalued so expert criteria could be needed and this knowledge will reinforce the selection of candidates prioritized. It is a key of the scheme and process of selection to have a good information base beforehand (literature, reports and databases). The higher and better the quality of the information, the more rigorous the selection of components will be. Experts and specialists examined each biodiversity component and judged their importance in the focal region to minimise the risk of benefiting certain charismatic taxa or a predetermined species as an indicator. Expert criteria and ecological importance require a thorough knowledge of species, habitats, ecological processes, and ecosystem disturbances, which is vital when selecting the appropriate relevant candidates for each focal region in which this methodology is to be implemented. This also allowed us to minimise the tendency to select rare species [67, 68] and helped us obtain more site-specific candidates [69]. The selection of the relevant candidates (species and habitats) to be included in a monitoring catalogue and their subsequent inclusion amongst the different priority indicators led to a third assessment of indicators via a multi-criteria analysis based on their specificity, generality, relevance, interactions with other organisms, and usefulness in management and decision-making.

Nevertheless, we placed special emphasis on the indicator selection for trophic networks and used a qualitative method strictly based on preselected indicators (species, habitats, and ecosystem disturbances) and their interrelations in the study area. All these interactions and relationships are depicted in a diagram (see State 3 of Methods, Fig 1), which provides information about the number of directly and indirectly related indicators. The integration and assessment of trophic networks–from primary production to predators–is a key component in the elaboration, selection, and monitoring of indicators [70, 71]. At the same time, an understanding of trophic networks is key for predicting trends and anticipating the conservationist or adaptive measures required in ecosystems [72]. Our method enables us to combine local results with the exploration of questions at broader geographical scales [73, 74].

This practical methodology for selecting indicators can be applied simply by conservation managers and nature stakeholders in local areas everywhere and networks such as the Barcelona Provincial Council's Network of Natural Parks (https://parcs.diba.cat/). Nevertheless, it is also useful as a management tool since it allows for better-informed and more cost-effective decision-making in a particular site [29, 49, 50]. Therefore, besides providing information about the current situation of ecosystems, the selected indicators can be used as decision criteria and as early warning signals of change in a specific region. This is especially relevant and necessary in a constantly changing world affected by natural and anthropic impacts [75]. It also allows us to formulate and implement biodiversity conservation strategies in changing landscapes through the use of a comprehensive and structured information system. Finally, these indicators also offer metrics for ecosystem status and provide interpretable information regarding changes [76].

### 4.1. Practical considerations before implementing indicators

After the selection of the definitive indicators, the following logical factors should be considered before implementing any monitoring program at the local scale: (1) an appropriate

selection of monitoring points with which to monitor the relevant candidates; (2) the generation of user-friendly protocols; and (3) an assessment of the ability of the protocols to obtain the required information. Indicator monitoring must be planned in the long term since complex dynamics and relationships in ecosystems may potentially affect decision quality. Long-term data can tackle questions not easily addressed in the short term [74] and are an excellent means of understanding how ecosystem disturbances impact ecosystem functioning and species dynamics. For example, Failing and Gregory [77] defend prescribed burns and replicating natural disturbance regimes to achieve long-term sustainable levels of biodiversity. Krebs [46] also highlights the need to have monitoring programs that report continuous information and data, having hundreds of years as time frame, and remarks on the importance of maintaining extended discussion of the monitoring problem, what should be monitored, and what the costs will be.

Monitoring scales should also be defined according to the sampling needs of each of the biological indicators. If a parameter cannot be adequately sampled, its usefulness for monitoring is greatly reduced [50]. Here three different scales are proposed: point-scale, plot-scale, and macro-scale (or landscape). If it is to be statistically robust, the monitoring scale has to offer the possibility of replicas, which do not suppose huge efforts, which can be used in the subsequent comparison.

1. Point-scale monitoring is intended for site-specific indicators that require very accurate, specific, and concrete monitoring due to the scarcity or specificity of the habitat they occupy (e.g. threatened and rare flora), or to their linear distribution in space (e.g. riparian communities).

2. Plot-scale monitoring within a homogeneous habitat with continuity at a fine-scale reveals the variations and trends occurring in different indicators. The monitoring of the selected indicators in the same plot will thus establish interactions according to each habitat (e.g. small-mammals, birds, habitat structure, trophic networks, decomposition, etc.). To obtain robust results, a minimum of four 100×100m Permanent Monitoring Plots are required in each selected habitat.

3. Macro-scale (or landscape) monitoring corresponds to cases in which indicators require wide-ranging monitoring (a whole protected area, a mountain range, a slope, a cliff, etc.) given the high mobility, ubiquity, or impact of the indicators (e.g. afforestation, predators, etc.). This type of monitoring will follow protocols and dimensions dependent on the specific indicator to be followed; however, they have in common the fact that they will all cover considerable parts of the region.

User-friendly monitoring protocols should be standardised and adapted to the selected indicators in consensus with experts and bearing in mind the initiatives and monitoring processes that are already being carried out at local, national, and international levels. Indicators can form part of more than one level of monitoring, and some may require more than one protocol since certain species, habitat components, and ecological processes necessarily demand specific methodologies.

## Supporting information

**S1 Appendix. Data sources.**
(DOCX)

**S1 Table. List and selection values of species.** List of the Individual species indicators of the Natural Park of Sant Llorenç del Munt i l'Obac following the criterion of achieving a value of

the Relevance index. The list includes the taxon community or aggrupation (bold font), the accepted name of the species, the Relevance index for each species, the average for the community or aggrupation (bold font), and the four subsections to extract this index: the degree of threat (scoring ranges from 0 to 3), the monitoring programs set (0 to 1), the ecological interest (0 to 2) and the expert or taxon specialist criterion (0 to 2).
(DOCX)

**S2 Table. List of aggregated habitats.** List of the 15 aggregated habitats (by similarity and type of plant formation) that collect in them the 96 habitats present in the Natural Park of Sant Llorenç del Munt i l'Obac. This list shows, the name of the aggregated habitat, the CORINE codes that are grouped in each aggregated habitat, and the description of the aggregated habitat.
(DOCX)

**S3 Table. Selection values of habitats.** List of the valued habitats (aggregation by similarity and type of plant formation) of the Natural Park of Sant Llorenç del Munt i l'Obac following the criterion of achieving a value of the Relevance index greater than 4 points on a total of 8. The Relevance index of each selected grouping habitat corresponds to the average of the Relevance indexes of the different habitats that make up the group according to the selection criteria (relevance $\geq$ 4), being the sum of the four subsections to extract this index: degree of threat (scoring ranges from 0 to 2), the ecological interest (0 to 2), representativeness (0 to 2) and the expert or specialist criterion (0 to 2). Representativeness corresponds to the number of selected species that appear in the grouping habitat relativized on 2, being 380 the maximum of species assessed.
(DOCX)

**S4 Table. List and selection values of ecosystem disturbances.** List of the valued ecosystem disturbances on Natural Park of Sant Llorenç del Munt i l'Obac following the criterion of achieving a value of the Relevance index greater than 2.5 points on a total of 5 (in bold). The list includes the ecosystem disturbances, the relevance index, and the three subsections to extract this index: the representativeness (scoring ranges from 0 to 2), the affected habitat (0 to 2), and the expert or specialist criterion (0 to 1). Representativeness corresponds to the number of selected species affected by the ecosystem disturbances relativized on 2, being 380 the maximum of species assessed. Habitat grouping is affected by the ecosystem disturbances relative to 2, with 11 being the maximum number of habitats assessed.
(DOCX)

**S1 Fig. Selection of ecological processes.** The scheme used to select ecological processes in of the Natural Park of Sant Llorenç del Munt i l'Obac by a diagram of interactions between primary producers, food sources, primary consumers, secondary consumers, predators, decomposers, and abiotic and anthropogenic external ecosystem disturbances, such as climatology, perturbations, pollution, forest management, etc.
(DOCX)

## Acknowledgments

This work has been carried out through the Biodiversity Monitoring Centre of Mediterranean Mountains (CMBMM) thanks a convention of Diputació de Barcelona and Universitat de Barcelona. So we are grateful to C. Castell, J. Puigdollers, R. Espinach, J. Bellapart, M.A. Palacio, L. Saurí, J. Padrós, J. Barber, A. Miño, D. Pons, J. Elias, D. Espriu and J. Garcia for their support. We would also like to thank all the experts consulted X. Santos, T. Hernández-Matías, M. López-Roig, J. Serra-Cobo, J. Alberch, G. Llorente, E. Pujol-Buxó, E. Llop, E. Mateos, A. Serra, M. Arnedo, À. Rollán, S. Mañosa, S. Sabaté, T. Sauras-Yera, L. Sáez, P. Pons, C. Gómez, L.

Brotons, J. Camprodon, D. Villero, I. Torre, C. Stefanescu, S. Herrando, L. Comas, J. Retana, A. Giménez, V. Bros, T. Mampel, P. Fernández, J. Canals, D. Carrera, A. Peris and A. Fàbrega, for their help and support during the project, both at the institutional level and at expert or taxon specialist level. We are indebted to a two anonymous referees and the academic editor Ignasi Torre that improved the manuscript.

## Author Contributions

**Conceptualization:** Roger Puig-Gironès, Joan Real.

**Data curation:** Roger Puig-Gironès.

**Formal analysis:** Roger Puig-Gironès.

**Funding acquisition:** Joan Real.

**Methodology:** Roger Puig-Gironès, Joan Real.

**Project administration:** Joan Real.

**Resources:** Roger Puig-Gironès.

**Supervision:** Joan Real.

**Validation:** Roger Puig-Gironès, Joan Real.

**Visualization:** Roger Puig-Gironès.

**Writing – original draft:** Roger Puig-Gironès, Joan Real.

**Writing – review & editing:** Roger Puig-Gironès, Joan Real.

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
