## [Decision Letter · Decision Letter 0]

19 Nov 2021

PONE-D-21-29903A comprehensive but practical methodology for selecting biological indicators for long-term monitoring Methodology for selecting biological indicatorsPLOS ONE

Dear Dr. Puig-Gironès,

Thank you for submitting your manuscript to PLOS ONE. After careful consideration, we feel that it has merit but does not fully meet PLOS ONE’s publication criteria as it currently stands. Therefore, we invite you to submit a revised version of the manuscript that addresses the points raised during the review process.

We look forward to receiving your revised manuscript.

Kind regards,

Ignasi Torre

Academic Editor

PLOS ONE

Journal Requirements:

2. We note that Figure 3 in your submission contain map images which may be copyrighted. All PLOS content is published under the Creative Commons Attribution License (CC BY 4.0), which means that the manuscript, images, and Supporting Information files will be freely available online, and any third party is permitted to access, download, copy, distribute, and use these materials in any way, even commercially, with proper attribution. For these reasons, we cannot publish previously copyrighted maps or satellite images created using proprietary data, such as Google software (Google Maps, Street View, and Earth). For more information, see our copyright guidelines: http://journals.plos.org/plosone/s/licenses-and-copyright.

1. You may seek permission from the original copyright holder of Figure 3 to publish the content specifically under the CC BY 4.0 license.  

Reviewers' comments:

Reviewer's Responses to Questions

**Comments to the Author**

1. Is the manuscript technically sound, and do the data support the conclusions?

Reviewer #1: Yes

Reviewer #2: Yes

2. Has the statistical analysis been performed appropriately and rigorously? 

Reviewer #1: Yes

Reviewer #2: N/A

3. Have the authors made all data underlying the findings in their manuscript fully available?

Reviewer #1: Yes

Reviewer #2: Yes

4. Is the manuscript presented in an intelligible fashion and written in standard English?

Reviewer #1: Yes

Reviewer #2: Yes

5. Review Comments to the Author

Reviewer #1: Dear author (s),

Thanks a lot for your valuable info that you provided in your targeted study and really sounds great for all the stockholders in the fields of biological and environmental science. Only you need to rewrite the abstract to be more clear and insert your aims of study also.

Reviewer #2: In their manuscript „a comprehensive but practical methodology for selecting biological indicators for long-term monitoring” the authors present how they selected a set of indicators to monitor changes in biodiversity at different levels in a Mediterranean ecosystem. The selection of suitable indicators is an important aspect to set up an effective monitoring system. The comprehensive approach is of interest to researchers facing the same challenge. This makes the paper a timely and important contribution.

The paper is very readable. The tables and figures are informative and I found the supporting information good and well organized. In the legend of figure 3 (L290) BMX appears twice (BMX: mixed forests AND BMX deciduous forests).

The method presented in the manuscript is described by the authors as simple and practical. The steps in themselves are logical and easy to understand, however the exact execution is less clear. The authors build on a large amount of literature and prior research and their work flow includes the consultation of experts. Line 377 discusses the risk of undervaluing some components; line 392 the use of a qualitative method strictly based on preselected indicators. While the authors discussion of these points is relevant and comprehensible, they do contribute to the impression that the method is not as easy to implement as the authors initially imply. The pre-requirements and the time aspect could be discussed further, especially as the aim of the method is to make monitoring more manageable.

Some of the aspects of the methodology, specifically the step from biodiversity components to indicators, were not entirely clear to me. The methodology does not sufficiently distinguish between parameters (species, habitats, processes, factors) and the assessment methods (abundances, phenology etc.). For instance, line 233 gives butterflies as an example, but does not specify an actual parameter that is measured. On page 8 and 9, the authors set a threshold of >= 4 for their index. I was unsure how this threshold was set and of its implications for the selection of species. In Table 2, “ecological process” stood out to me as a criterion: the description could apply to all species. The methodology that allowed to group the 434 relevant components into 32 specific monitoring components requires more details.

Relevant components selection: I’m not sure about the grouping of species: while I understand that the % of alien species is of interest, they could also fit into the taxonomic groups. I think it would be more easily understandable to provide the numbers separately (classification into organism groups and a separate % for status). It might also be more easy to give species numbers rather than % - please check the numbers, as the % in the text do not always seem to match up with the number of species given in Annex 2.

6. PLOS authors have the option to publish the peer review history of their article (what does this mean?). If published, this will include your full peer review and any attached files.

Reviewer #1: **Yes: **Salwan Ali Abed

Reviewer #2: No

---

## [Author Response · Author response to Decision Letter 0]

31 Dec 2021

Reviewer #1: 

Thanks a lot for your valuable info that you provided in your targeted study and really sounds great for all the stockholders in the fields of biological and environmental science. Only you need to rewrite the abstract to be more clear and insert your aims of study also.

Many thanks for the review. We have tried to clarify our abstract.

Reviewer #2: 

In their manuscript „a comprehensive but practical methodology for selecting biological indicators for long-term monitoring” the authors present how they selected a set of indicators to monitor changes in biodiversity at different levels in a Mediterranean ecosystem. The selection of suitable indicators is an important aspect to set up an effective monitoring system. The comprehensive approach is of interest to researchers facing the same challenge. This makes the paper a timely and important contribution. The paper is very readable. The tables and figures are informative and I found the supporting information good and well organized. 

Many thanks for your point of view about our contribution to the selection of suitable indicators.

In the legend of figure 3 (L290) BMX appears twice (BMX: mixed forests AND BMX deciduous forests).

Thanks. We changed the abbreviation error.

The method presented in the manuscript is described by the authors as simple and practical. The steps in themselves are logical and easy to understand, however the exact execution is less clear. The authors build on a large amount of literature and prior research and their work flow includes the consultation of experts. Line 377 discusses the risk of undervaluing some components; line 392 the use of a qualitative method strictly based on preselected indicators. While the authors discussion of these points is relevant and comprehensible, they do contribute to the impression that the method is not as easy to implement as the authors initially imply. 

Thanks for the comments. To improve the comprehensibility of our process we have improved the explanation from the 125 to 133 lines (Previously lines 119 to 146) and we have modified figure 1. Furthermore, we have rewritten and reordered the methods, especially State 3, State 4 and the new redaction of Results (sections 3.2 and 3.3). 

The pre-requirements and the time aspect could be discussed further, especially as the aim of the method is to make monitoring more manageable.

Our method presents certain key steps that can compromise the good selection of indicators. Especially by the own intricacies of each taxon. For this reason, our method presents three evident prerequisites: 

• Make an exhaustive collection of information and especially the most difficult, unpublished reports.

• Having knowledgeable naturalists in the area.

• To be able to consult specialists.

Then, understanding your concern, we have made some adjustments into State 2 of M&M; 

New lines from 158 to 161: “An effective collection of prior information on the area to be applied, especially not published reports, is essential. If relevant prior information is not available, you need to back to State 1 to search for or generate relevant information”.

We also incorporate into the discussion, some new arguments about prerequisites (new lines from 394 to 399): 

“Some possible biases could arise from the exclusive use of the weight values in the Relevance Index as some potential candidates (e.g. plants and invertebrate species) would be undervalued so expert criteria could be needed and this knowledge will reinforce the selection of candidates prioritized. It is a key of the scheme and process of selection to have a good information base beforehand (literature, reports and databases). The higher and better the quality of the information, the more rigorous the selection of components will be.”

This complements other phrases that emphasize the expert importance in the introduction, e.g. new line from 86 to 90: “[...] the knowledge of experts or specialists in local taxa is essential since one aspect of biodiversity (species, habitats, ecological processes, and biotic, abiotic and anthropic problems) may affect a focal region differently and so require its indicator [6]. It is also important to assess which indicators are valid and informative for a region and which are redundant, overvalued or unnecessary”. 

Moreover, in methods, e.g. new lines from 198 to 200: "[...] the bibliography and expert criteria are essential (Table 2), and will help highlight species or habitats that are not present or not evaluated as threatened [...]".

Some of the aspects of the methodology, specifically the step from biodiversity components to indicators, were not entirely clear to me. The methodology does not sufficiently distinguish between parameters (species, habitats, processes, factors) and the assessment methods (abundances, phenology etc.). For instance, line 233 gives butterflies as an example, but does not specify an actual parameter that is measured. 

To facilitate the discrimination between parameters and method, we reformulate the new lines 228 to 244, of the State 4: Monitoring Catalogue section of the document:

“Once the different relevant components have been assessed and selected, a Monitoring Catalogue (Figure 1) can be established to act as a strategic guide, facilitate understanding, and identify which variables need to be sampled and measured to generate the biological indicators in an ecological sense and to avoid redundancies. To attain the list of relevant components, (1) we grouped components by taxonomic, physiognomic, and functional similarities (e.g. common birds, freshwater invertebrates, decomposers). Then, (2) classified the grouped relevant components in four ecological levels: species, habitat, ecological process, and ecosystem disturbances (see the case study below as a practical example). (3) For each group of relevant components we selected the adequate variables to be sampled, e.g., species presence and abundance, reproductive and survival taxes, habitat preferences, soil structure, and composition. Lastly, (4) incorporated into each level all the monitoring components whilst bearing in mind that each can be part of one or more levels depending on their role and interactions within the ecosystem (see State 3). For example, species abundance and richness of butterflies provide information about the species level and their community (indicators at species level), about the variation on ecosystem functional capabilities, e.g., as primary consumers (indicators at ecological process level), or climate change and afforestation effects (indicators at ecosystem disturbance level)”

On page 8 and 9, the authors set a threshold of >= 4 for their index. I was unsure how this threshold was set and of its implications for the selection of species. 

We understand the question raised. Here we tried to establish a criterion that was rigorous and, at the same time, did not rule out underrepresented components. For this purpose, the criterion was half of the Relevance index. Thus, this criterion was chosen to greatly reduce the number of initial components but, at the same time, to ensure that those assessed by the experts were not lost. 

We have incorporated the following explanation in new lines 198 to 193 of the State 3: The establishment of the relevant components of monitoring biodiversity selection for clarification:

“Because the threshold value needs to be sufficiently robust but at the same time to include potential under-detected components, we established as a criterion to maintain the components with a Relevance index equal to or above half of the maximum global potential value that included valuable ecosystem components, as we have verified in our previous tests (see Table 1).”

In Table 2, “ecological process” stood out to me as a criterion: the description could apply to all species. 

Here, the selection criterion corresponding to the description was applied to all species. Unfortunately, this information is not known in-depth for all species, especially invertebrates. 

The methodology that allowed to group the 434 relevant components into 32 specific monitoring components requires more details.

We have redefined the section State 4: Monitoring Catalogue section (new lines from 228 to 244) of the document for clarification, and wrote all grouped components: 

“Once the different relevant components have been assessed and selected, a Monitoring Catalogue (Figure 1) can be established to act as a strategic guide, facilitate understanding, and identify which variables need to be sampled and measured to generate the biological indicators in an ecological sense and to avoid redundancies. To attain the list of relevant components, (1) we grouped components by taxonomic, physiognomic, and functional similarities (e.g. common birds, freshwater invertebrates, decomposers). Then, (2) classified the grouped relevant components in four ecological levels: species, habitat, ecological process, and ecosystem disturbances (see the case study below as a practical example). (3) For each group of relevant components we selected the adequate variables to be sampled, e.g., species presence and abundance, reproductive and survival taxes, habitat preferences, soil structure, and composition. Lastly, (4) incorporated into each level all the monitoring components whilst bearing in mind that each can be part of one or more levels depending on their role and interactions within the ecosystem (see State 3). For example, species abundance and richness of butterflies provide information about the species level and their community respectively, so in the latter case indicate the variation on ecosystem functional capabilities (e.g. as primary consumers), climate change or afforestation effects. ”

Ant the 3.3. Monitoring catalogue and Priority indicators selection (new lines from 355 to 373):

“To constitute the Monitoring catalogue, we grouped the 434 Relevant components independently by the level of species, habitats, ecological processes, and ecosystem disturbances by similarities. In the case of 408 species were grouped by taxonomic, physiognomic, and functional similarities into 17 groups of species (threatened plant species, xeric gastropods, granivorous ants, decomposers, orthoptera, butterflies, freshwater macroinvertebrates, freshwater fishes, amphibians, reptiles, common birds, raptors, bats, small mammals, common medium-size preys, ungulates and carnivores). A total of 96 habitats were grouped in 11 habitat relevant categories (Table 4). Then, the habitat category resulted in two monitoring parameters (habitat structure and composition), which will be carried out in the 11 selected relevant habitats. Moreover, it is in these habitats where the specific transversal monitoring of species, processes and disturbances will be developed. Ecological processes were grouped by ecological functions in 6, decomposition, primary production, consumers, predators, zoocory, and pollination. Ecosystem disturbances were maintained in 8 (Table 4), climatic change, wildfires, afforestation, freshwater alterations, alien species, exploitation of natural resources, forests pests, human frequentation, human infrastructures). So resulted in a total of 32 specific Monitoring Components. Finally, to obtain the Priority Indicators we used the quantitative Priority Index, and 27 of these Monitoring Components were selected to monitor the biodiversity in the PNSLL (Table 5).”

Relevant components selection: I’m not sure about the grouping of species: while I understand that the % of alien species is of interest, they could also fit into the taxonomic groups. 

Although the reviewer is right (alien species could be grouped into the respective taxonomic groups), in previous discussions with managers and experts, we decided to give a category for relevant alien species, as an independent indicator. However, they are noted in surveys when autochthonous species (e.g. vegetation) are noted, i.e., they do not need a special survey. However, Then, in this case, as in others, the same species indicates on more than one level: as an indicator of species status, as well as an indicator of the landscape and ecosystem disturbances (as biological invasion). 

I think it would be more easily understandable to provide the numbers separately (classification into organism groups and a separate % for status). It might also be more easy to give species numbers rather than % - please check the numbers, as the % in the text do not always seem to match up with the number of species given in Annex 2.

Thanks, we reviewed the numbers.

Reviewer suggestion was incorporated on 3.2. Relevant components selection section (new lines from 314 to 323):

“The relevant components were obtained through a Relevance Index which was calculated for all 3,226 assessed species (Table 4) of which 406 (12.6%) were selected as Relevant Components (S1 Table): plants contributed with 69 species (17% - of which 10 species were threatened species, 13 pteridophytes, 41 vascular plants, and five alien species), 191 species (47%) were invertebrates (of which 26 species were Odonata, 12 Gastropoda, 43 decomposers, 22 Formicidae, 30 Orthoptera; 36 Lepidoptera, ten other invertebrates of special interest, seven plague species and five were alien species), while 146 species (36%) were vertebrates (of which tree species were freshwater fish, seven amphibians, seven reptiles, 13 raptors, 70 common birds, 21 bats, seven small-mammals, two common medium-sized preys, two ungulates, five carnivores and nine were alien species)”

Thanks, we reviewed the numbers.

---

## [Editor Report · Decision Letter 1]

11 Feb 2022

PONE-D-21-29903R1A comprehensive but practical methodology for selecting biological indicators for long-term monitoringPLOS ONE

Dear Dr. Puig-Gironès,

Thank you for submitting your manuscript to PLOS ONE. After careful consideration, we feel that it has merit but does not fully meet PLOS ONE’s publication criteria as it currently stands. Therefore, we invite you to submit a revised version of the manuscript that addresses the points raised during the review process.

I acknowledge the huge effort performed to summarise and synthetize all the information recorded, in such an easy way to be processed by the readers. However, there are still some points to be addressed prior to acceptance:

Line 174: include references for “monitoring programs”* and “expert criterion”

The scale values assigned to each criterion need to be further justified. Why “Threat” is in a scale 0-3, “Ecological Interest” in a scale 0-2, and “presence on monitoring programs” in scale 0-1? Why assigning more weight to Threat than to Ecological Interest? This represents a subjective starting point decided by authors.

In supporting Table, I see some data that not match my knowledge on the species selected. Some small mammals having monitoring programs, scored “0”. Also, expert criterion is low for species with reduced populations (M.glareolus, E.quercinus), so it’s disappointing. This needs to be corrected or justified. I don’t know if such differences are affecting to other taxonomic groups. This will affect final scores, and need to be recalculated, if necessary.

Also raised by reviewers…Why a threshold is set in the mean, >=4 ? A rank criterion, ordering the species from 0-8, will be better since between 5-8 there are significant differences of suitability that could help selecting in particular situations (ex. limited resources). If one wants to select the most relevant indicators, I will choose those having 8 scores.

Expert criteria seem a rather subjective selection criterion. Indeed, experts use similar criteria for selecting indicators regarding threat and ecological interest of the species, so it could be a redundant variable. Furthermore, a “0” indicates that experts consider the species irrelevant, or there is no information available?

Table 1. why “z”? It should correspond to “f”. Please, justify.

Table 2. Correct “Specie” to “Species”

*Torre, I., López-Baucells, A., Stefanescu, C., Freixas, L., Flaquer, C., Bartrina, C., Coronado, A., López-Bosch, D., Mas, M., Míguez, S., Muñoz, J., Páramo, F., Puig-Montserrat, X., Tuneu-Corral, C., Ubach, A., Arrizabalaga, A., 2021. Concurrent Butterfly, Bat and Small Mammal Monitoring Programmes Using Citizen Science in Catalonia (NE Spain): A Historical Review and Future Directions. Diversity 13, 454. doi:10.3390/D13090454

Please consider whether your article meets PLOS ONE criteria for manuscripts that describe new methods. Specifically, these reports must meet the criteria of utility, validation, and availability, which are described in detail at http://journals.plos.org/plosone/s/submission-guidelines#loc-methods-software-databases-and-tools.

We look forward to receiving your revised manuscript.

Kind regards,

Ignasi Torre

Academic Editor

PLOS ONE
---

## [Author Response · Author response to Decision Letter 1]

24 Feb 2022

Academic Editor:

I acknowledge the huge effort performed to summarise and synthetize all the information recorded, in such an easy way to be processed by the readers. However, there are still some points to be addressed prior to acceptance:

Line 174: include references for “monitoring programs”* and “expert criterion”.

*Torre, I., López-Baucells, A., Stefanescu, C., Freixas, L., Flaquer, C., Bartrina, C., Coronado, A., López-Bosch, D., Mas, M., Míguez, S., Muñoz, J., Páramo, F., Puig-Montserrat, X., Tuneu-Corral, C., Ubach, A., Arrizabalaga, A., 2021. Concurrent Butterfly, Bat and Small Mammal Monitoring Programmes Using Citizen Science in Catalonia (NE Spain): A Historical Review and Future Directions. Diversity 13, 454. doi:10.3390/D13090454.

Thanks for the reference, we have added the paper to the references. On the other hand, we have included “Duelli P, Obrist MK. Biodiversity indicators: the choice of values and measures. Agriculture, ecosystems & environment. 2003; 98(1):87-98. doi: 10.1016/S0167-8809(03)00072-0” for the expert criterion.

The scale values assigned to each criterion need to be further justified. Why “Threat” is in a scale 0-3, “Ecological Interest” in a scale 0-2, and “presence on monitoring programs” in scale 0-1? Why assigning more weight to Threat than to Ecological Interest? This represents a subjective starting point decided by authors.

Thanks for your comment. Degree of threat and Ecological Interest were to be given equal weight in the selection process. In the specific case of species, initially the "presence on monitoring programs" count as part of Ecological interest, as they correspond to common species (commonly detected species in monitoring plans). However, to avoid the misunderstandings and subjectivity that this implies, we have placed this category ("presence on monitoring programs") within Ecological interest. In this way, Ecological interest and Degree of threat have equal weight (from 0 to 3) in the final model (see Table 1, Table 2, and Table S1).

This is well explained in the new lines 76-90, 197-204 and 406-422 of the manuscript. However, we have also modified the following sentence in the new line 201-202 of the manuscript, to facilitate understanding:

“The ecological interest contributes to minimize the weight of these threatened and singular component, which do not necessarily reflect the spatial and temporal trends of biodiversity [23, 55] but, rather, respond to the local conservation status”.

In supporting Table, I see some data that not match my knowledge on the species selected. Some small mammals having monitoring programs, scored “0”. Also, expert criterion is low for species with reduced populations (M.glareolus, E.quercinus), so it’s disappointing. This needs to be corrected or justified. I don’t know if such differences are affecting to other taxonomic groups. This will affect final scores, and need to be recalculated, if necessary.

In the first case, the criterion was to assign a “1” if the species was detected previously in a specific monitoring program. Due to E.quercinus was captured by the monitoring program in 2018 and M.glareolus has the potential to be captured by SEMICE monitoring program; have updated the assessment of the two species. However, we have not updated the rest of the species since in the monitoring that has been carried out since 2018 (the database used here was last updated in 2017) in the natural park there have not been any new records, see Puig-Gironès R. 2020 Catàleg de Flora Vascular, Fauna Invertebrada i Fauna Vertebrada del Parc Natural de Sant Llorenç del Munt i l’Obac. Diputació de Barcelona, Barcelona.

The document presented is not intended to be an immovable document, and it is proposed to be re-evaluated in a couple of years (2024 or 2025). Therefore, we have decided not to modify the evaluation of the expert criterion since this revaluation will be made for the totality of the species. Experts and specialists were consulted to evaluate the ecological and conservation interest of each species (see acknowledgements for the list of the consulted experts), and they examined each biodiversity component and judged their importance in the focal region to minimise the risk of benefiting certain charismatic taxa or a predetermined species as an indicator.

Also raised by reviewers…Why a threshold is set in the mean, >=4 ? A rank criterion, ordering the species from 0-8, will be better since between 5-8 there are significant differences of suitability that could help selecting in particular situations (ex. limited resources). If one wants to select the most relevant indicators, I will choose those having 8 scores.

Thanks for the comment. We understand the reasoning of the editor and reviewers. However, the aim of this paper, as detailed in the introduction and discussion, is to be a balanced method (see again, the new lines 76 – 90, 197-204 and 406-422). 

Given the dissimilarities in the knowledge and subsequent assessment of the species, to put a very high threshold as 8 entail biased indicators to more visible, detectable and studied species. For example, selecting those species with a Relevance index of 8, eight of the ten species of threatened and endemic flora would not be chosen, such as: Arenaria conimbricensis subsp. conimbricensis, Arenaria fontqueri or Erodium glandulosum, nor the endemic gastropods like Montserratina bofilliana, Xerocrassa montserratensis or Abida secale subsp. bofilli. 

On the other hand, if the threshold were ≥ 7, then 57 species would be selected, of which: 8 would be plants (14%), 5 invertebrates (8.8%) and 44 vertebrates (77.2%) so at first instance represents a high biased frequency compared to listed plants (32%), invertebrates (61%) and vertebrates (7%) in the Sant Llorenç del Munt Natural Park (see Puig-Gironès R. 2020 Catàleg de Flora Vascular, Fauna Invertebrada i Fauna Vertebrada del Parc Natural de Sant Llorenç del Munt i l’Obac. Diputació de Barcelona, Barcelona). 

These enormous differences are also maintained in the threshold ≥ 6 and 5. Therefore, we set the threshold at ≥ 4, since this limit allowed us to have reasonable percentages between plants, invertebrate and vertebrate species to follow as indicator.

Expert criteria seem a rather subjective selection criterion. Indeed, experts use similar criteria for selecting indicators regarding threat and ecological interest of the species, so it could be a redundant variable. Furthermore, a “0” indicates that experts consider the species irrelevant, or there is no information available?

Our main objective was a collaborative design process to enhance the acceptance of diverse values and prioritizations embedded in sustainability assessments. This emphasis on the process would make assessments more transparent, transformative and enduring. Even so, in certain species, habitats or processes there may be some redundancy, but we believe that the expert criterion based on the local scale minimized this effect. Furthermore, given the lack of monitoring systems or the lack of catalogues of endangered species (in the case of Catalonia we do not have a Catalogue of Endangered Fauna), expert judgement fills this gap.

On the other hand, expert criterion is necessary to assess which indicators are valid and informative for a region. As detailed in Table 2: “the criterion of external expert or taxon specialist grants a value of 2 to the species, habitats, and/or ecosystem disturbances with high importance and relevance in the context of the study area, and 1 to species, habitats, and/or ecosystem disturbances with relative importance in the study area”. 

Table 1. why “z”? It should correspond to “f”. Please, justify.

Thanks to finding the error, we have changed 'z' to 'f'.

Table 2. Correct “Specie” to “Species”

Thanks, we have fixed the error.

---

## [Editor Report · Decision Letter 2]

28 Feb 2022

A comprehensive but practical methodology for selecting biological indicators for long-term monitoring

PONE-D-21-29903R2

Dear Dr. Puig-Gironès,

We’re pleased to inform you that your manuscript has been judged scientifically suitable for publication and will be formally accepted for publication once it meets all outstanding technical requirements.

Kind regards,

Ignasi Torre

Academic Editor

PLOS ONE
---

## [Editor Report · Acceptance letter]

4 Mar 2022

PONE-D-21-29903R2 

A comprehensive but practical methodology for selecting biological indicators for long-term monitoring 

Dear Dr. Puig-Gironès:

I'm pleased to inform you that your manuscript has been deemed suitable for publication in PLOS ONE. Congratulations! Your manuscript is now with our production department. 

Kind regards, 

on behalf of

Dr. Ignasi Torre 

Academic Editor

PLOS ONE